

# Risk factors and predictive model for renal outcomes in autoimmune membranous nephropathy with and without acute kidney injury: a retrospective cohort study

Zhenzhou Li[1,2,3,*], Liyan Yang[1,2,3,*], Linxia Wei[1,2], Mengjie Weng[1,2,3], Jiaqun Lin[1,2,3], Yi Chen[1,2,3], Binbin Fu[1,2,3], Guifen Li[1,2,3], Caiming Chen[1,2,3], Yanfang Xu[1,2,3], Jianxin Wan[1,2,3] and Jiong Cui[1,2,3]

[1] Department of Nephrology, Blood Purification Research Center, the First Affiliated Hospital, Fujian Medical University, Fuzhou, Fujian, China
[2] Fujian Clinical Research Center for Metabolic Chronic Kidney Disease, the First Affiliated Hospital, Fujian Medical University, Fuzhou, Fujian, China
[3] Department of Nephrology, National Regional Medical Center, Binhai Campus of the First Affiliated Hospital, Fujian Medical University, Fuzhou, Fujian, China
* These authors contributed equally to this work.

Corresponding authors
Jianxin Wan, wanjx@fjmu.edu.cn
Jiong Cui, cuijiong@fjmu.edu.cn

## ABSTRACT

**Objective:** This study aimed to delineate the risk factors affecting renal outcomes in autoimmune membranous nephropathy (aMN) with or without acute kidney injury (AKI) and develop a predictive model.

**Methods:** This retrospective cohort study included 441 patients with biopsy-confirmed aMN from the First Affiliated Hospital of Fujian Medical University (January 2010 to March 2023). Patients were grouped based on the presence of AKI and followed up until a renal endpoint event (progression to end-stage renal disease, initiation of dialysis, or either a >40% decline in estimated glomerular filtration rate from baseline or a doubling of serum creatinine levels from baseline, both sustained for ≥3 months) or study endpoint (March 2024). Clinicopathological and renal outcomes were collected and analyzed. Risk factors for renal endpoints were identified *via* Cox regression analyses, and a nomogram was constructed. Model performance was evaluated using the C-index, time-dependent receiver operating characteristic (Time-ROC) curves, calibration curves, and decision curve analysis (DCA). Kaplan–Meier survival curves compared renal survival between AKI subgroups.

**Results:** Among 441 patients, 109 (24.72%) experienced AKI. Renal endpoint events occurred in 40.4% of the AKI group and 4.5% of the non-AKI group. Multivariate Cox regression identified AKI (HR = 7.298, $P < 0.001$), triglycerides (HR = 1.140, $P = 0.002$), serum creatinine (HR = 1.008, $P = 0.012$), hematuria (HR = 2.246, $P = 0.040$), and kidney anti-M-type phospholipase A2 receptor staining 4+ (HR = 2.473, $P = 0.003$) as independent risk factors, while serum C3 (HR = 0.082, $P < 0.001$) was an independent protective factor. The nomogram had a C-index of 0.845 ($P < 0.001$), with Time-ROC AUCs of 0.92, 0.81, 0.83, and 0.87 for 3 to 6 years, respectively. Calibration plots revealed good consistency between the predicted and actual probabilities. DCA indicated that the nomogram had potential clinical utility.

Kaplan–Meier analysis showed lower cumulative renal survival in patients with AKI ($P < 0.001$).

**Conclusions:** The risk factor model suggests that renal outcomes in patients with aMN can be predicted. Early assessment and management targeting these identified risk factors could help delay renal function decline in these patients.

## INTRODUCTION

Membranous nephropathy (MN) is one of the most common pathological types of adult nephrotic syndrome (NS). Traditionally, MN can be classified as idiopathic membranous nephropathy (IMN) and secondary membranous nephropathy (sMN), with IMN diagnosed after excluding secondary causes such as hepatitis B, systemic autoimmune diseases, malignancies, drugs, and toxins (*Sharma et al., 2022*). Recent advances have redefined idiopathic MN as an autoimmune disorder mediated by circulating autoantibodies targeting multiple podocyte antigens, including phospholipase A2 receptor (PLA2R), thrombospondin type-1 domain containing 7A (THSD7A), high-temperature requirement A serine peptidase 1 (HTRA1), Semaphorin 3B, and Netrin G1, among others (*Avasare, Andeen & Beck, 2024*; *Sethi et al., 2023*). Given that approximately 80–90% of previously 'idiopathic' cases are now attributable to these antigen-antibody interactions, experts advocate replacing the term IMN with autoimmune MN (aMN) to emphasize its autoimmune-driven nature (*Cattran & Brenchley, 2017*; *Hoxha, Reinhard & Stahl, 2022*). Consistent with this paradigm shift, our study employs aMN as the classification for MN cases without identifiable secondary etiologies.

NS is the primary manifestation of aMN, and its major complications include acute kidney injury (AKI), infections, and thromboembolism (*Agrawal et al., 2018*). Patients with aMN have a significantly higher risk of experiencing cardiovascular events even in the early stages of the disease (*Lee et al., 2016*). As aMN progresses, renal function deteriorates progressively, as evidenced by a decline in the estimated glomerular filtration rate (eGFR), with approximately 10–40% of cases eventually progressing to end-stage renal disease (ESRD) within 10–20 years (*Shiiki et al., 2004*). ESRD further increases the risk of cardiovascular events and mortality, leading to a poorer prognosis. Therefore, early identification of risk factors for prevention to delay the progression of renal dysfunction in aMN is crucial for reducing the risks of ESRD, cardiovascular disease, and mortality, thereby improving long-term outcomes.

Patients with aMN can experience AKI during the course of the disease. Our previous study (*Li et al., 2023*) found that among 187 patients with aMN, 46 (24.6%) developed AKI, which increased the risk of adverse renal outcomes (progression to ESRD or a need for renal replacement therapy). Another recent study (*Chen et al., 2021*) indicated that among 434 patients with aMN and NS, 124 (28.6%) developed AKI, which was identified as an

independent risk factor for the progression of chronic kidney disease (CKD) in this cohort. However, research on the risk factors and predictive analysis of renal outcomes in patients with aMN according to the presence of AKI is limited. Therefore, this study explored renal outcomes in patients with aMN with or without AKI and constructed a prognostic predictive model to enable individualized management and early assessment to delay the progression of renal function deterioration and optimize renal outcomes.

## MATERIALS AND METHODS

### Study population

This single-center, retrospective cohort study included patients who underwent renal biopsy and received a pathological diagnosis of MN at the Nephrology Department of the First Affiliated Hospital of Fujian Medical University (Fuzhou, China) between January 2010 and March 2023. The inclusion criteria were as follows: age of 18–90 years, eGFR $\geq$ 15 mL/min/1.73 m$^2$, and not currently pregnant. Patients with sMN caused by systemic lupus erythematosus, hepatitis B virus infection, Sjögren's syndrome, hyperthyroidism, and other diseases; comorbid diseases such as malignancy, diabetic nephropathy, hypertensive nephropathy, and focal segmental glomerulosclerosis; and incomplete clinical data were excluded. All included patients underwent at least 1 year of follow-up, with follow-up concluding in March 2024. Basic clinical and pathological data were collected during hospitalization for renal biopsy. This study was conducted in accordance with the Declaration of Helsinki, and the protocol was approved by the Medical Research and Clinical Technology Application Ethics Committee of the First Affiliated Hospital of Fujian Medical University (MTCA, ECFAH of FMU [2015] 084-2). Informed written consent was obtained from all patients.

### Variable descriptions

Demographic characteristics examined in the study included age, sex, BMI, systolic blood pressure (SBP), and diastolic blood pressure (DBP). AKI, diabetes, and hypertension were monitored as comorbidities. Regarding treatment, the use of ACE inhibitors or angiotensin receptor blockers, steroids, and immunosuppressants (including FK506, mycophenolate mofetil, cyclosporine, cyclophosphamide, and monoclonal antibody biological agents) was also recorded. In blood tests, baseline serum creatinine (SCr), baseline eGFR, serum uric acid, triglycerides, total cholesterol, low-density lipoprotein (LDL) cholesterol, serum albumin, hemoglobin, fibrinogen, and serum complement C3 levels were monitored. Urine test indicators included 24-h urinary protein (24 h UPr) and hematuria (urinary sediment microscopy showing red blood cells $\geq$ 3/HP). Serum anti-M-type phospholipase A2 receptor (PLA2R) antibody levels were classified using four semi-quantitative tiers: Tier 1 (1:10), Tier 2 (1:32), Tier 3 (1:100), and Tier 4 ($\geq$1:320).

In the histopathological evaluation of renal biopsy specimens, the following parameters were assessed: glomerulosclerosis; crescent formation; tubular atrophy; interstitial inflammation; interstitial fibrosis; and the deposition of IgG, C3, IgG4, and PLA2R antigen in the kidneys. The chronicity index and tubulointerstitial lesions were determined using the method described by *Austin et al. (1994)*. The intensity of immunofluorescence

staining for IgG, C3, IgG4 and PLA2R antigen deposits in renal tissue was graded on a semi-quantitative scale (range: 1 to 4+), with a grade of 1+ or higher considered positive.

Serum creatinine content was measured using the Cobas 8000 (Roche Diagnostics, Mannheim, Germany) enzymatic method, and eGFR was calculated using the CKD-EPI equation (*Levey et al., 2009*). Triglyceride levels were measured using the Cobas 8000 (Roche Diagnostics, Mannheim, Germany) oxidase method, serum C3 levels were measured using the IMMAGE 800 (Beckman Coulter, Brea, CA, USA) immunoturbidimetric method, and 24 h UPr levels were determined using the Cobas 8000 (Roche Diagnostics, Mannheim, Germany) turbidimetric method. Serum PLA2R antibodies were detected using the Sprinter XL (Euroimmun, Lübeck, Germany) indirect immunofluorescence method.

## Definitions

The renal study endpoint was a composite of progression to ESRD or initiation of dialysis, a decrease in eGFR of >40% from baseline (≥3 months), or a doubling of serum creatinine levels from baseline (≥3 months). Patients were followed through medical record inquiries and telephone interviews until March 2024 or until the occurrence of a renal composite endpoint event. Data from patients who were lost to follow-up, those who died, and those who completed the full follow-up period without experiencing a renal composite endpoint event were censored.

AKI was diagnosed if any one of the following conditions was met during follow-up: an increase in SCr of at least 26.5 µmol/L within 48 h; an increase in SCr of at least 1.5-fold above the baseline value within 7 days; or hourly urine output of <0.5 mL/kg for >6 h. Additionally, some patients had elevated SCr levels on the first day of admission, followed by a decline. In such cases, the baseline SCr was assumed to be lower than that on the first day based on the subsequent clinical course. Thus, based on the 2012 KDIGO Clinical Practice Guidelines for AKI, these patients were also diagnosed with AKI (*Khwaja, 2012*). AKI stage 1 was defined as an increase in SCr to 1.5–1.9-fold above the baseline level or an increase in SCr of ≥26.5 µmol/L. AKI stage 2 was defined as an increase in SCr to 2.0–2.9-fold above the baseline. AKI stage 3 was defined as an increase in SCr to ≥3.0-fold higher than the baseline, an increase in SCr of ≥354 µmol/L with an acute increase of at least 44 µmol/L, or the initiation of renal replacement therapy (*Khwaja, 2012*). Comorbid AKI described the occurrence of at least one episode of AKI at the time of diagnosis or during follow-up in patients with aMN.

Meanwhile, diabetes and hypertension were identified through ICD-10 coding in electronic medical records.

## Statistical analysis

For continuous variables, the Kolmogorov–Smirnov test was used to assess normality. Normally distributed data were expressed as the mean ± standard deviation, and comparisons between groups were conducted using Student's *t*-test. Non-normally distributed data were presented as the median and interquartile range (IQR), and group comparisons were performed using the Mann–Whitney U test. Categorical variables were

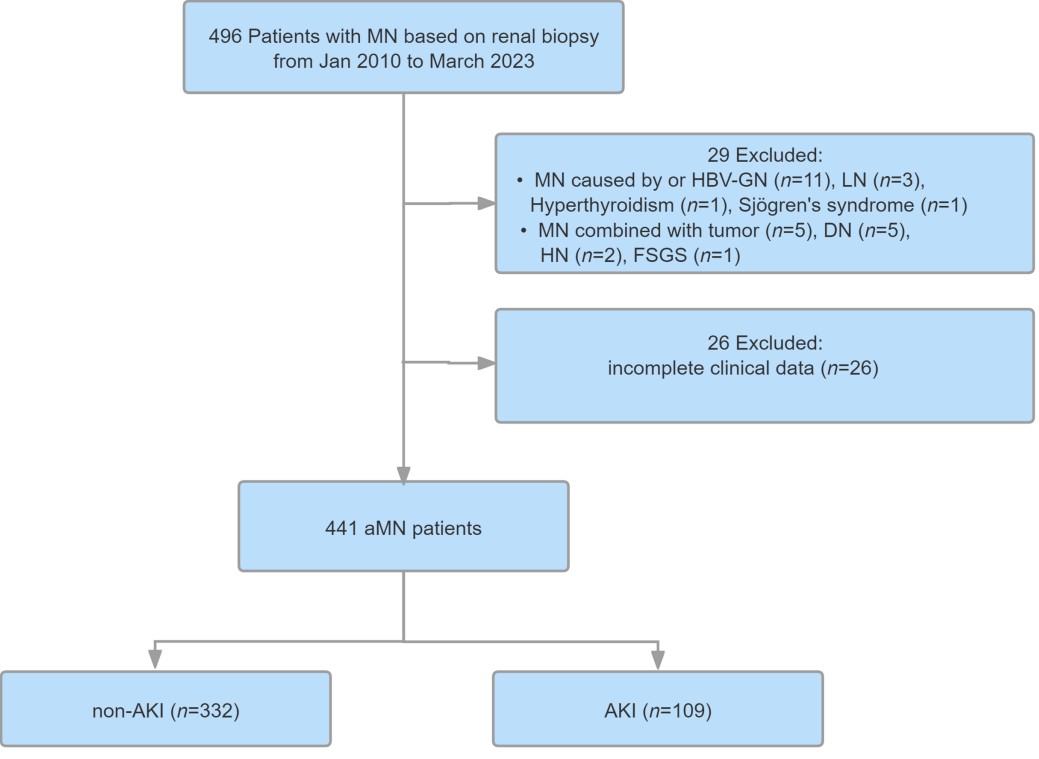

**Figure 1 Study flowchart.** MN, membranous nephropathy; aMN, autoimmune membranous nephropathy; LN, lupus nephritis; HBV-GN, hepatitis B virus-associated glomerular nephritis; DN, diabetic nephropathy; AKI, acute kidney injury.

expressed as counts and percentages, and comparisons between groups were performed using the chi-squared test. Kaplan–Meier survival curves were plotted, and survival analysis was conducted using the log rank test. Univariate Cox regression analysis was conducted to identify variables affecting renal endpoints in patients with aMN, and variables significant at $P < 0.05$ were included in the multivariate Cox proportional hazards model. To assess the robustness of the model, bootstrap internal validation was performed using 1,000 iterations of resampling to compute the C-index and evaluate calibration curves. The final model was used to construct a nomogram for predicting renal endpoints in patients with aMN. Differences were considered statistically significant at $P < 0.05$. All analyses were performed using R software version 4.2.3 (R Core Team, Vienna, Austria) and SPSS software version 26.0 (IBM, Armonk, NY, USA).

# RESULTS

## Study population characteristics

A total of 496 patients with MN were enrolled in this study. After screening, 441 of these were diagnosed with aMN and included in the Cox analysis model (Fig. 1). The median age of the enrolled patients was 54 years (IQR = 43–63), and 63.9% were male. The median follow-up period was 43 months (IQR = 26–63), and total follow-up time was 1,655 person-years. Among the 441 patients, 109 (24.72%) experienced AKI, including 82 (75.23%), 21 (19.27%), and six patients (5.50%) categorized into AKI stages 1, 2, and 3,

**Table 1 Basic clinical and pathological characteristics of the study population (continuous variables).**

| Variable | All ($n$ = 441) | Without AKI ($n$ = 332) | With AKI ($n$ = 109) | $t$/Z | P |
|---|---|---|---|---|---|
| Follow-up duration (months) | 43.00 (26.00, 63.00) | 42.00 (25.25, 62.00) | 53.00 (26.00, 68.50) | −1.141 | 0.254 |
| **Demographic characteristics** | | | | | |
| Age, years | 54.00 (43.00, 63.00) | 53.00 (42.00, 63.00) | 55.00 (48.00, 63.00) | −1.887 | 0.059 |
| SBP, mmHg | 130.00 (117.50, 144.00) | 128 (117.00, 143.00) | 134 (122.50, 146.50) | −2.777 | 0.005** |
| DBP, mmHg | 80.00 (73.00, 88.00) | 79.00 (73.00, 87.00) | 81.00 (72.00, 91.50) | −1.347 | 0.178 |
| BMI, kg/m$^2$ | 23.98 (21.37, 25.86) | 23.82 (21.22, 25.70) | 24.20 (21.96, 26.33) | −1.708 | 0.088 |
| **Clinical characteristics** | | | | | |
| Baseline SCr, μmol/L | 70.60 (56.00, 87.00) | 66.90 (54.00, 83.98) | 80.00 (64.55, 100.00) | −4.527 | <0.001*** |
| Baseline eGFR (mL/min/1.73 m$^2$) | 107.46 ± 38.46 | 112.03 ± 38.95 | 93.54 ± 33.42 | 4.447 | <0.001*** |
| Uric acid, μmol/L | 381.31 ± 101.48 | 371.82 ± 97.10 | 410.25 ± 109.26 | −3.474 | 0.001** |
| Triglycerides, mmol/L | 2.01 (1.41, 2.89) | 2.01 (1.35, 2.87) | 2.01 (1.49, 2.98) | −0.249 | 0.803 |
| Total cholesterol, mmol/L | 7.17 (5.55, 8.63) | 7.14 (5.49, 8.61) | 7.45 (5.97, 8.87) | −0.657 | 0.511 |
| LDL cholesterol, mmol/L | 4.69 (3.25, 6.39) | 4.66 (3.21, 6.39) | 4.90 (3.41, 6.46) | −0.498 | 0.618 |
| Serum albumin, g/L | 24.76 (21.65, 29.10) | 24.76 (21.90, 29.90) | 23.10 (20.80, 27.30) | −2.68 | 0.007** |
| Hemoglobin, g/L | 129.64 ± 18.15 | 130.10 ± 18.41 | 128.22 ± 17.35 | 0.940 | 0.348 |
| Fibrinogen, g/L | 4.27 (3.35, 5.32) | 4.20 (3.33, 5.25) | 4.56 (3.63, 5.58) | −1.654 | 0.098 |
| 24h UPr, g/24 h | 4.08 (2.27, 6.81) | 3.99 (2.03, 6.62) | 4.43 (2.91, 8.16) | −2.643 | 0.008** |
| Serum C3, g/L | 0.92 (0.79, 1.01) | 0.92 (0.80, 1.01) | 0.92 (0.75, 1.00) | −0.866 | 0.387 |
| **Renal pathology score** | | | | | |
| TIL | 3 (3, 3) | 3 (3, 3) | 3 (3, 3) | −1.825 | 0.068 |
| Chronicity index | 2 (2, 3) | 2 (2, 3) | 3 (2, 3) | −3.367 | 0.001** |

Notes:
Data are expressed as the mean ± SD (normally distributed) or median and interquartile range (non-normally distributed). Abbreviations: AKI, acute kidney injury; SBP, systolic blood pressure; DBP, diastolic blood pressure; BMI, body mass index; SCr, serum creatinine; eGFR, estimated glomerular filtration rate; LDL, low-density lipoprotein; 24 h UPr, 24-h urinary protein; TIL, tubular interstitial lesion. Student's $t$-test was used to compare normally distributed variables, whereas the Mann–Whitney U test was used to compare non-normally distributed variables. (Two-tailed).
** $P$ < 0.01.
*** $P$ < 0.001.

respectively. The remaining 332 patients (75.28%) did not experience AKI. Of the 441 study participants, 59 experienced renal endpoint events, including 44 patients (40.36%) in the AKI group and 15 patients (4.52%) in the non-AKI group.

Compared with the non-AKI group, the AKI group had a higher proportion of male patients, higher rates of hypertension and steroid use, and higher SBP. The AKI group also exhibited higher baseline levels of serum uric acid and SCr, lower serum albumin levels and eGFR, higher 24 h UPr levels, higher rates of glomerulosclerosis and crescent formation, and a higher chronicity index. Additionally, the AKI group had a higher rate of immunosuppressant use and a higher incidence of renal adverse events (all $P$ < 0.05). No other variables differed between the groups. Detailed data are presented in Tables 1 and 2.

### Univariate Cox regression analysis
Univariate Cox regression analysis, we identified the following variables as significant risk predictive factors: AKI, baseline SCr, baseline eGFR, triglycerides, hematuria, DBP, high PLA2R expression in the kidneys (4+), chronicity index, and glomerulosclerosis. Among

**Table 2** Basic clinical and pathological characteristics of the study population (categorical variables).

| Variable | All ($n = 441$) | Without AKI ($n = 332$) | With AKI ($n = 109$) | $\chi^2$ | $P$ |
|---|---|---|---|---|---|
| **Demographic characteristics** | | | | | |
| Male sex, $n$ (%) | 282 (63.9%) | 200 (60.2%) | 82 (75.2%) | 7.996 | 0.005** |
| **Clinical characteristics** | | | | | |
| Hypertension, $n$ (%) | 225 (51.0%) | 155 (46.7%) | 70 (64.2%) | 10.095 | 0.001** |
| Diabetes, $n$ (%) | 65 (14.7%) | 45 (13.6%) | 20 (18.3%) | 1.501 | 0.221 |
| Hematuria, $n$ (%) | 142 (32.2%) | 106 (31.9%) | 36 (33.0%) | 0.045 | 0.831 |
| Serum PLA2R antibody positive, $n$ (%) | | | | | |
| Negative | 127 (28.8%) | 101 (30.4%) | 26 (23.9%) | 1.727 | 0.189 |
| 1:10 | 30 (6.8%) | 24 (7.2%) | 6 (5.5%) | 0.385 | 0.535 |
| 1:32 | 82 (18.6%) | 58 (17.5%) | 24 (22.0%) | 1.122 | 0.290 |
| 1:100 | 140 (31.7%) | 106 (31.9%) | 34 (31.2%) | 0.020 | 0.886 |
| 1:320 | 62 (14.1%) | 43 (13.0%) | 19 (17.4%) | 1.363 | 0.243 |
| Therapy status | | | | | |
| ACEI/ARB, $n$ (%) | 380 (86.2%) | 280 (84.3%) | 100 (91.7%) | 3.776 | 0.052 |
| Glucocorticoids, $n$ (%) | 287 (65.1%) | 205 (61.7%) | 82 (75.2%) | 6.563 | 0.010* |
| Immunosuppressants, $n$ (%) | 282 (63.9%) | 194 (58.4%) | 88 (80.7%) | 17.700 | <0.001*** |
| **Renal pathologies** | | | | | |
| Glomerular sclerosis, $n$ (%) | 203 (46.0%) | 141 (42.5%) | 62 (56.9%) | 6.860 | 0.009** |
| Crescent formation $n$ (%) | 22 (5.0%) | 12 (3.6%) | 10 (9.2%) | 5.352 | 0.021* |
| Tubular atrophy, $n$ (%) | 425 (96.4%) | 320 (96.4%) | 105 (96.3%) | 0.001 | 0.979 |
| Interstitial inflammatory infiltration, $n$ (%) | 435 (98.6%) | 326 (98.2%) | 109 (100%) | | 0.344 |
| Interstitial fibrosis, $n$ (%) | 420 (95.2%) | 316 (95.2%) | 104 (95.4%) | 0.010 | 0.921 |
| Kidney IgG deposition, $n$ (%) | 410 (93.0%) | 307 (92.5%) | 103 (94.5%) | 0.515 | 0.473 |
| Kidney C3 deposition, $n$ (%) | 328 (74.4%) | 240 (72.3%) | 88 (80.7%) | 3.071 | 0.080 |
| Kidney IgG4 deposition, $n$ (%) | 363 (82.3%) | 271 (81.6%) | 92 (84.4%) | 0.435 | 0.510 |
| Kidney PLA2R staining, $n$ (%) | | | | | |
| No staining | 87 (19.7%) | 66 (19.9%) | 21 (19.3%) | 0.020 | 0.889 |
| PLA2R + | 60 (13.6%) | 44 (13.3%) | 16 (14.7%) | 0.142 | 0.706 |
| PLA2R ++ | 71 (16.1%) | 55 (16.6%) | 16 (14.7%) | 0.216 | 0.642 |
| PLA2R +++ | 140 (31.7%) | 104 (31.3%) | 36 (33.0%) | 0.110 | 0.740 |
| PLA2R ++++ | 83 (18.8%) | 63 (19.0%) | 20 (18.3%) | 0.021 | 0.884 |
| **Outcome variable** | | | | | |
| Composite endpoint, $n$ (%) | 59 (13.4%) | 15 (4.5%) | 44 (40.4%) | 91.000 | <0.001*** |

Notes:

Data are expressed as numbers (percentages). Abbreviations: AKI, acute kidney injury; PLA2R, phospholipase A2 receptor; ACEI, ACE inhibitor; ARB, angiotensin receptor blocker. The chi-squared test was used to compare categorical variables.

* $P < 0.05$.
** $P < 0.01$.
*** $P < 0.001$.

these, AKI, hematuria, and PLA2R (4+) exhibited strong associations with clinical outcomes. Additionally, serum C3 was identified as a significant protective predictive factor, with increased levels closely associated with a lower risk of renal endpoint events (all $P < 0.05$, Table 3).

**Table 3  Univariate Cox regression analysis of factors affecting renal endpoint events in patients with aMN.**

| Variable | Beta | HR | CI | P |
|---|---|---|---|---|
| **Demographic characteristics** | | | | |
| Male sex | 0.330 | 1.390 | [0.773–2.502] | 0.272 |
| Age | 0.012 | 1.012 | [0.995–1.032] | 0.242 |
| SBP | 0.012 | 1.012 | [0.999–1.024] | 0.063 |
| DBP | 0.030 | 1.030 | [1.010–1.051] | 0.004** |
| BMI | 0.038 | 1.038 | [0.972–1.109] | 0.267 |
| **Clinical characteristics** | | | | |
| Baseline SCr | 0.012 | 1.012 | [1.007–1.017] | <0.001*** |
| Baseline eGFR | −0.014 | 0.986 | [0.979–0.994] | <0.001*** |
| Triglycerides | 0.077 | 1.080 | [1.011–1.153] | 0.023* |
| Uric acid | 0.001 | 1.001 | [0.999–1.004] | 0.355 |
| Total cholesterol | 0.025 | 1.025 | [0.927–1.134] | 0.627 |
| LDL cholesterol | 0.036 | 1.037 | [0.925–1.162] | 0.532 |
| Serum albumin | −0.038 | 0.962 | [0.919–1.007] | 0.099 |
| Hemoglobin | −0.002 | 0.998 | [0.983–1.012] | 0.746 |
| Fibrinogen | −0.045 | 0.956 | [0.823–1.111] | 0.559 |
| 24 h UPr | 0.002 | 1.002 | [0.933–1.075] | 0.966 |
| Serum C3 | −2.364 | 0.094 | [0.024–0.372] | 0.001** |
| Serum PLA2R antibody | 0.056 | 1.058 | [0.580–1.929] | 0.855 |
| Serum PLA2R antibody 1:10 | −0.325 | 0.722 | [0.225–2.318] | 0.585 |
| Serum PLA2R antibody 1:32 | −0.190 | 0.827 | [0.419–1.634] | 0.585 |
| Serum PLA2R antibody 1:100 | 0.138 | 1.148 | [0.678–1.945] | 0.608 |
| Serum PLA2R antibody 1:320 | 0.208 | 1.231 | [0.603–2.515] | 0.569 |
| AKI | 2.044 | 7.719 | [4.287–13.898] | <0.001*** |
| Hematuria | 1.008 | 2.741 | [1.291–5.820] | 0.009** |
| Hypertension | 0.298 | 1.348 | [0.805–2.257] | 0.256 |
| Diabetes | 0.213 | 1.237 | [0.625–2.449] | 0.541 |
| Therapy status | | | | |
| ACEI/ARB | 0.293 | 1.341 | [0.536–3.356] | 0.531 |
| Glucocorticoids | 0.081 | 1.084 | [0.622–1.890] | 0.776 |
| Immunosuppressants | 0.372 | 1.451 | [0.806–2.613] | 0.215 |
| **Renal pathologies** | | | | |
| TIL | 0.130 | 1.139 | [0.909–1.428] | 0.259 |
| Chronicity index | 0.248 | 1.282 | [1.004–1.637] | 0.046* |
| Glomerular sclerosis | 0.515 | 1.674 | [1.001–2.798] | 0.049* |
| Crescent formation | 0.397 | 1.487 | [0.464–4.767] | 0.505 |
| Tubular atrophy | −0.600 | 0.549 | [0.198–1.522] | 0.249 |
| Interstitial inflammatory infiltration | −0.139 | 0.870 | [0.120–6.293] | 0.890 |
| Interstitial fibrosis | 0.498 | 1.646 | [0.400–6.765] | 0.490 |
| Kidney IgG deposition | −0.604 | 0.547 | [0.218–1.372] | 0.198 |
| Kidney C3 deposition | −0.108 | 0.898 | [0.491–1.641] | 0.726 |

| Variable | Beta | HR | CI | P |
|---|---|---|---|---|
| Kidney IgG4 deposition | −0.502 | 0.605 | [0.326–1.122] | 0.111 |
| Kidney PLA2R staining | −0.155 | 0.856 | [0.462–1.587] | 0.622 |
| Kidney PLA2R + | 0.043 | 1.044 | [0.512–2.129] | 0.906 |
| Kidney PLA2R ++ | −0.803 | 0.448 | [0.179–1.121] | 0.086 |
| Kidney PLA2R +++ | −0.372 | 0.689 | [0.383–1.239] | 0.214 |
| Kidney PLA2R ++++ | 0.796 | 2.216 | [1.252–3.921] | 0.006** |

**Notes:**
Abbreviations: SBP, systolic blood pressure; DBP, diastolic blood pressure; BMI, body mass index; SCr, serum creatinine; eGFR, estimated glomerular filtration rate; LDL, low-density lipoprotein; 24 h UPr, 24-h urinary protein, AKI, acute kidney injury; ACEI, ACE inhibitor; ARB, angiotensin receptor blocker; TIL, tubular interstitial lesion; PLA2R, phospholipase A2 receptor; HR, hazard ratio; CI, confidence interval.
* $P < 0.05$.
** $P < 0.01$.
*** $P < 0.001$.

## Multivariate Cox regression analysis

The significant variables in univariate analysis were included in the multivariate Cox regression model. The results illustrated that AKI (HR = 7.298, 95% confidence interval [CI] = 4.018–13.255, $P < 0.001$), triglycerides (HR = 1.140, 95% CI [1.049–1.238], $P = 0.002$), baseline SCr (HR = 1.008, 95% CI [1.002–1.014], $P = 0.012$), hematuria (HR = 2.246, 95% CI [1.039–4.852], $P = 0.040$), and kidney PLA2R staining 4+ (HR = 2.473, 95% CI [1.363–4.486], $P = 0.003$) were independent risk factors for renal endpoint events in patients with aMN, whereas serum C3 (HR = 0.082, 95% CI [0.020–0.342], $P < 0.001$) was an independent protective factor (Table 4). Our study indicated that the incidence of renal endpoint events in patients with aMN and AKI was 7.298-fold higher (95% CI [4.018–13.255]) than that in their counterparts without AKI.

## Construction and evaluation of the prognostic nomogram

Based on the variables with significant prognostic value for renal endpoint events in patients with aMN identified in multivariate Cox regression analysis, a nomogram model was developed to predict survival probabilities at 3–6 years (Fig. 2).

Internal validation of the nomogram model yielded a C-index of 0.845 (95% CI [0.792–0.898], $P < 0.001$). The predictive performance (discrimination) of the model was further evaluated using the area under the time-dependent receiver operating characteristic curve (AUC). The results indicated that the AUC was 0.92 (95% CI [0.873–0.977]) at 3 years, 0.81 (95% CI [0.717–0.906]) at 4 years, 0.83 (95% CI [0.745–0.911]) at 5 years, and 0.87 (95% CI [0.799–0.933]) at 6 years (Fig. 3). This suggests that the model has good discriminatory ability and predictive performance. The model was calibrated by plotting calibration curves using 1000 bootstrap resamples, as illustrated in Figs. 4A–4D. The calibration curves demonstrated strong concordance between the predicted and the actual renal survival probability at 3–6 years, suggesting good calibration (consistency) of the model. The clinical applicability of the model at different time points (3, 4, 5, and 6 years) was evaluated *via* decision curve analysis (DCA). As shown in

**Table 4 Multivariate Cox regression analysis of factors affecting renal endpoint events in patients with aMN.**

| Variable | Beta | HR [95% CI] | P |
|---|---|---|---|
| AKI | 1.988 | 7.298 [4.018–13.255] | <0.001*** |
| Triglycerides | 0.131 | 1.140 [1.049–1.238] | 0.002** |
| Baseline SCr | 0.008 | 1.008 [1.002–1.014] | 0.012* |
| Serum C3 | −2.497 | 0.082 [0.020–0.342] | <0.001*** |
| Hematuria | 0.809 | 2.246 [1.039–4.852] | 0.040* |
| Kidney PLA2R staining 4+ | 0.905 | 2.473 [1.363–4.486] | 0.003** |

Notes:
Abbreviations: AKI, acute kidney injury; SCr, serum creatinine; PLA2R, phospholipase A2 receptor; CI, confidence interval.
* $P < 0.05$.
** $P < 0.01$.
*** $P < 0.001$.

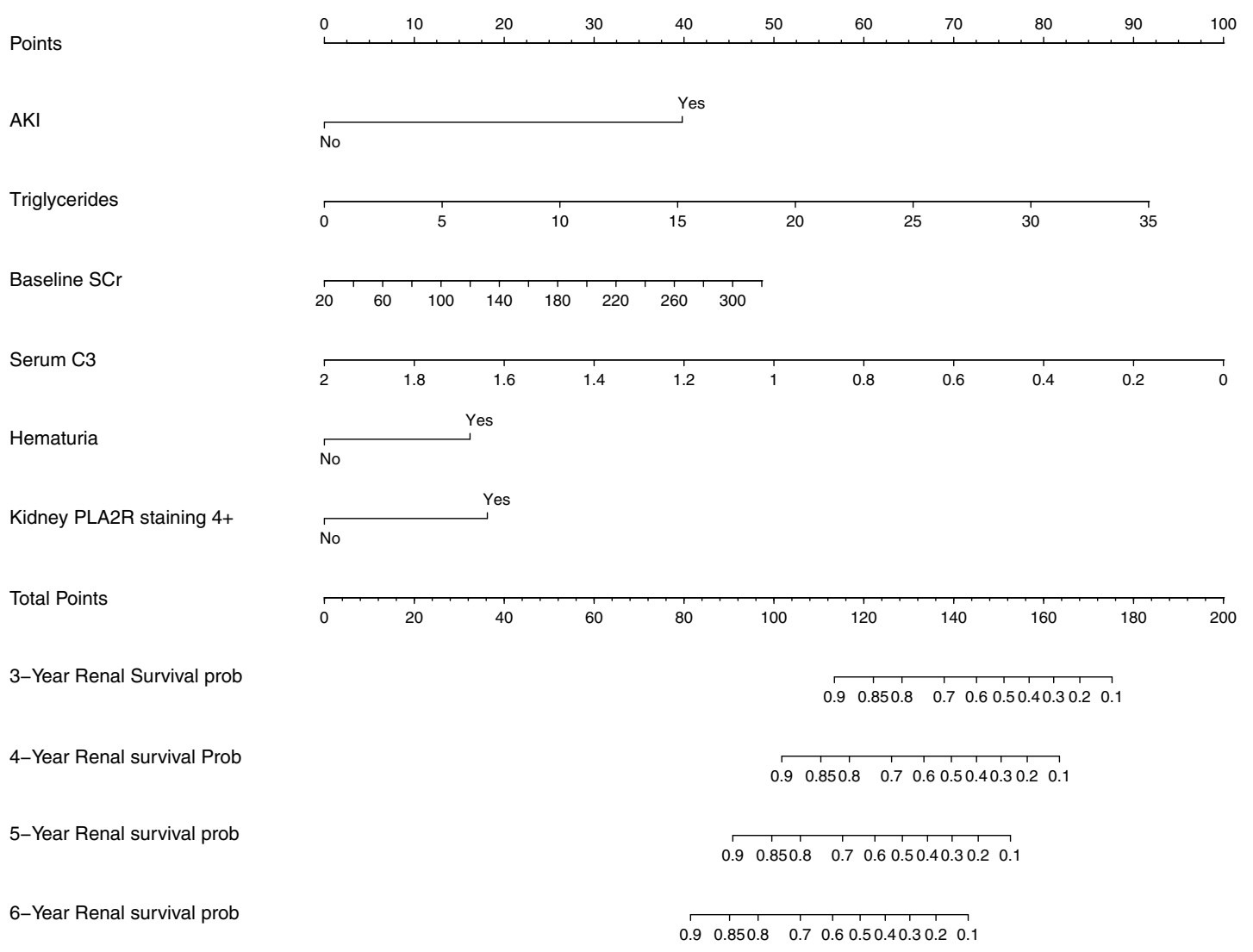

**Figure 2 Prognostic nomogram for predicting renal endpoint events in patients with aMN over a 3–6-year period.**

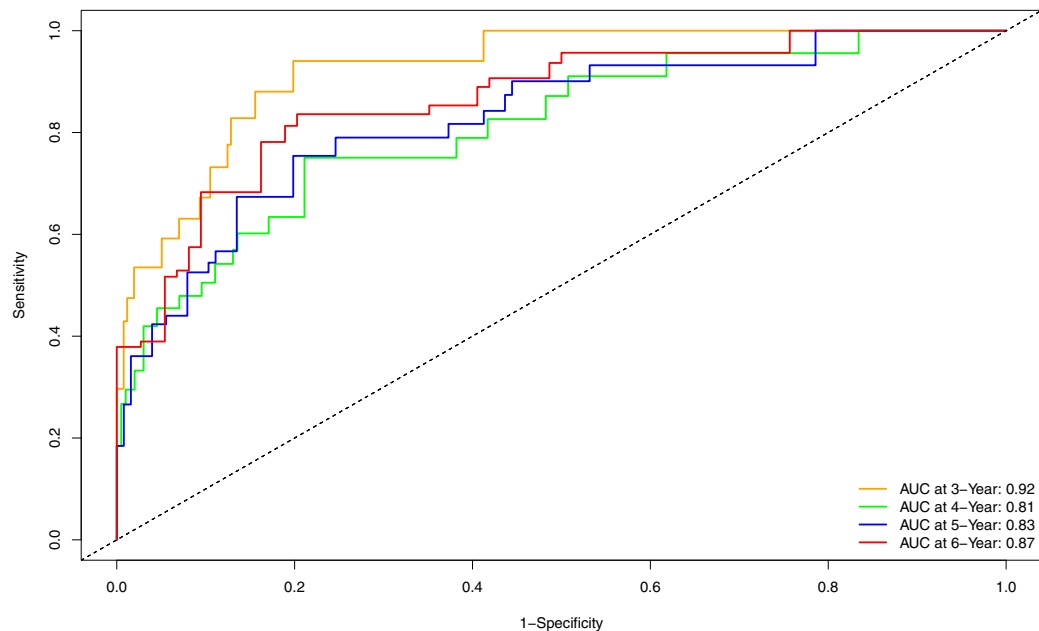

**Figure 3 Time-dependent ROC curve and AUC for predicting renal endpoint events in patients with aMN over a 3–6-year period.**

Figs. 5A–5D, the DCA curves indicated that the nomogram model provided more clinical benefits for predicting renal adverse events in patients with aMN when the threshold probability (probability of renal endpoint events) was 0.01–0.35 at 3 years, 0.01–0.50 at 4 years, 0.01–0.55 at 5 years, and 0.01–0.60 at 6 years.

Overall, these evaluation results indicate that the nomogram demonstrated high discrimination and good calibration in internal validation and provided significant clinical benefits at specific threshold probabilities, demonstrating overall good predictive performance.

## Kaplan–Meier survival curve analysis

We analyzed the renal survival rates of subgroups stratified by the presence of AKI in patients with aMN using Kaplan–Meier survival curves. The analysis revealed that the cumulative renal survival rate was significantly lower in the AKI group compared to the non-AKI group ($P < 0.001$, Fig. 6).

## DISCUSSION

The incidence of aMN has gradually increased in China. According to a study in southern China (*Zheng et al., 2022*), aMN comprises approximately 37.49% of all cases of primary glomerular disease, and its incidence has gradually increased from 2010 to 2018. Following treatment for aMN, some patients remain in remission (*Xu et al., 2020*). while others may experience a decline in renal function that progresses to CKD (*Couser, 2017*). Additionally, some patients may progress to renal failure (*Xu et al., 2020*) or ESRD and require dialysis (*Couser, 2017*; *Xu et al., 2020*). Therefore, identifying patients with aMN at high risk for disease progression and managing potential risk factors is crucial for nephrologists.

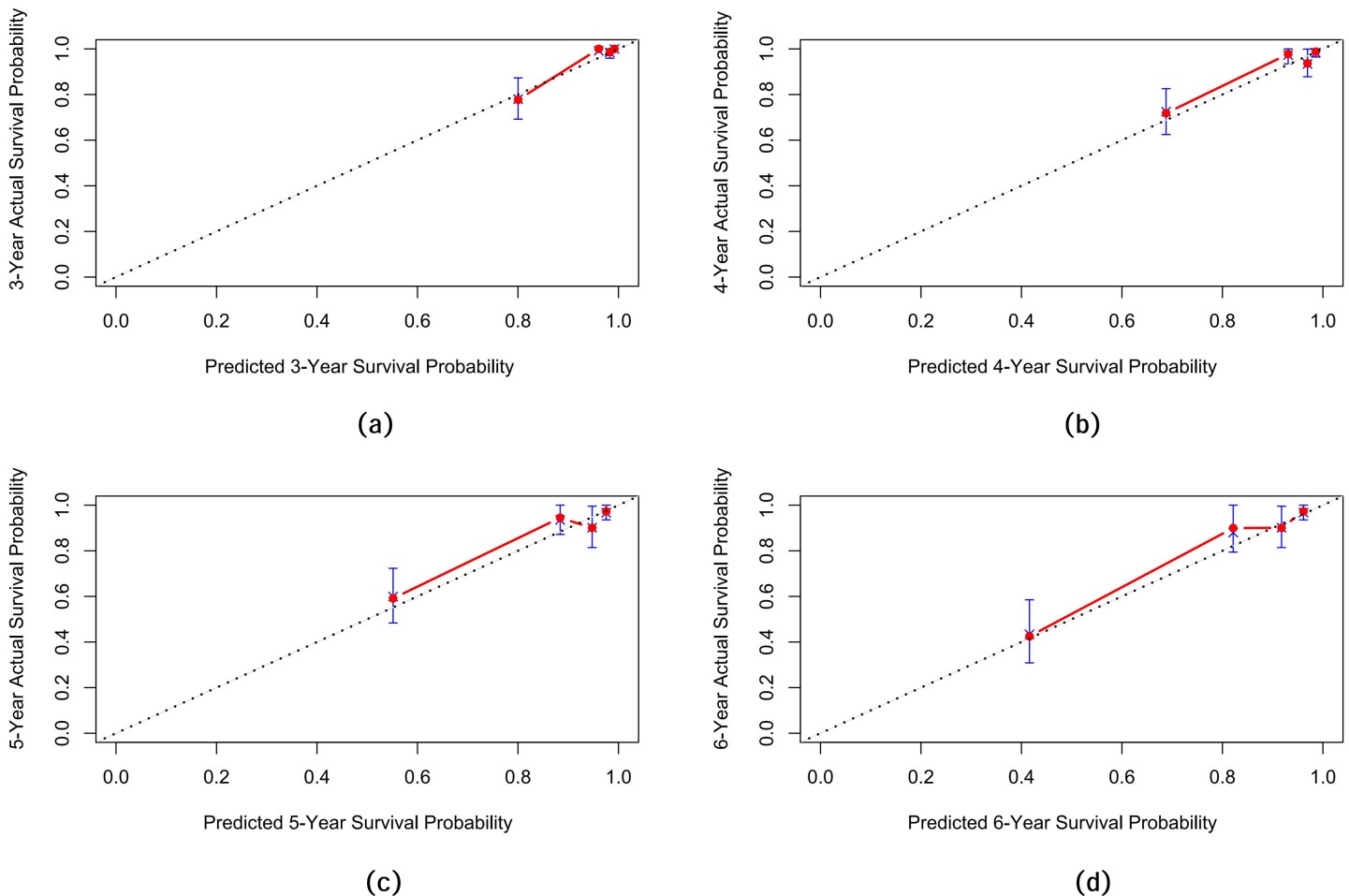

**Figure 4 Calibration curves of the predictive model for renal survival in patients with aMN at different time points.** (A) 3 years. (B) 4 years. (C) 5 years. (D) 6 years.

The renal outcome in patients with aMN is influenced by various factors. Studies have demonstrated that severe proteinuria (*Chen et al., 2019*), reduced baseline eGFR (*Zuo et al., 2013*), age ≥ 60 years, low serum albumin concentrations, and severe tubulointerstitial damage (*Yin et al., 2020*) are associated with poor renal prognosis in aMN. AKI is a relatively common complications of aMN that can lead to a poor prognosis. Most previous studies have focused on the outcomes of AKI in patients with nephrotic syndrome and minimal change disease (MCD) (*Fenton, Smith & Hewins, 2018*; *Meyrier & Niaudet, 2018*), but few studies have investigated the impact of AKI on renal outcomes specifically in aMN. Furthermore, there is a lack of reliable models predicting renal outcomes in patients with aMN and AKI. Therefore, based on previous research (*Li et al., 2023*), this study integrated clinical indicators, laboratory tests, and renal pathology to construct a nomogram predictive model for renal prognosis in aMN. This model was designed to facilitate the early identification of poor renal outcomes in patients with aMN and AKI, providing a basis for clinical diagnosis.

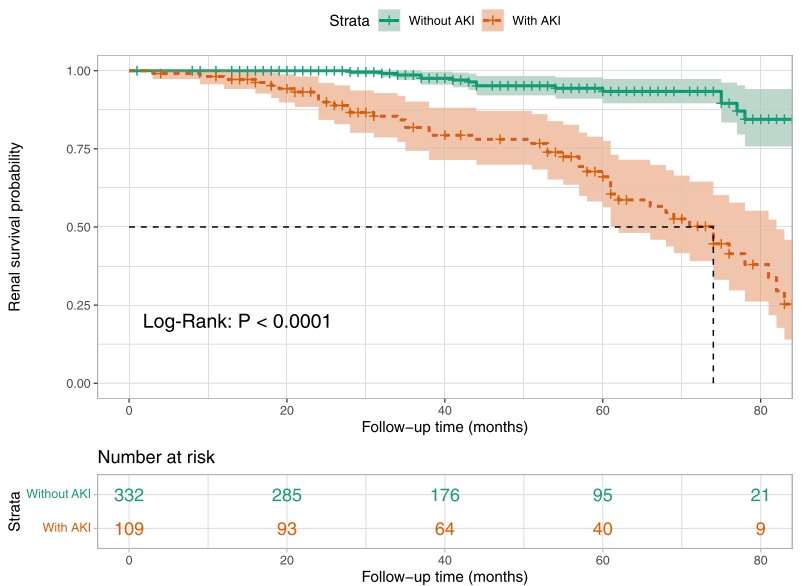

**Figure 5 Decision curve analysis of the predictive model for renal survival in patients with aMN at different time points.** (A) 3 years. (B) 4 years. (C) 5 years. (D) 6 years.

**Figure 6 Kaplan–Meier renal survival plot for groups stratified by the presence of AKI.**

Following AKI, patients may experience worsening renal dysfunction, significantly increasing the risk of CKD, ESRD, and even mortality (*See et al., 2019*; *Sykes et al., 2019*). However, there are limited reports on the renal prognosis of patients with aMN and AKI. In this study, we expanded the sample size and extended the follow-up period based on previous research to observe the occurrence of renal events. Even though 75.23% of patients with AKI were classified as AKI stage 1, the incidence of renal endpoint events was 7.298-fold higher in patients with aMN and AKI compared to those without AKI. This finding was consistent with a previous study (*See et al., 2019*), but the hazard ratio observed in our study was greater, suggesting that AKI in aMN was more detrimental than that in the general population.

Studies investigating the direct association of renal PLA2R with adverse renal events in patients with aMN are limited. In this study, strong renal PLA2R staining (4+) was identified as a major risk factor for worsening renal dysfunction in patients with aMN. PLA2R is a transmembrane protein located on podocytes and serves as the primary target antigen in aMN. Strong PLA2R staining in renal tissue is significantly associated with podocyte damage and proteinuria, highlighting its critical role in the pathogenesis of aMN (*Bobart et al., 2019*). Although peripheral blood PLA2R antibodies correlate with the severity of aMN and can assist in diagnosis without a kidney biopsy, our findings suggest that PLA2R staining in renal tissue is a more direct and reliable marker for assessing disease severity and predicting outcomes in the context of kidney biopsy. These findings regard PLA2R as independent risk factors for renal dysfunction. However, disease progression in aMN is multifactorial, and lipid metabolism disorders, particularly elevated serum triglycerides, have also been implicated in kidney function loss.

This study identified serum triglycerides as an independent risk factor for worsening renal dysfunction in patients with aMN. Clinical studies have demonstrated a correlation between abnormal triglyceride levels and renal damage in aMN. For instance, a single-center, retrospective cohort study involving 495 patients with aMN indicated that increased triglyceride levels are associated with tubular atrophy, as shown by logistic regression analysis (odds ratio = 1.025, $P = 0.011$) (*Dong et al., 2022*). Elevated triglycerides can lead to lipid accumulation in renal tubules, resulting in lipotoxicity and mediating tubular atrophy. The underlying mechanisms include inflammatory responses, oxidative stress, endoplasmic reticulum stress, apoptosis of tubular epithelial cells, and tubulointerstitial fibrosis, ultimately exacerbating renal dysfunction in aMN patients (*Guan et al., 2024*). However, despite our study identified statistical significance for triglycerides as a risk factor (HR = 1.080, CI [1.011–1.153], $P = 0.023$), the confidence interval was close to 1, suggesting that its clinical relevance should be interpreted with caution.

Serum complement C3, synthesized by the liver, is a globulin and a key component of the immune system, playing a core role in defending against microorganisms (*Copenhaver et al., 2020*). Research suggests that the classical, lectin, and alternative pathways of complement activation are all be involved in the pathogenesis of MN, mediating podocyte damage and local renal inflammation through mechanisms such as activation of the complement cascade, formation of the membrane attack complex, and binding of the C3

cleavage fragment C3a to C3aR on podocytes (*Gao, Cui & Zhao, 2022*). This process is often accompanied by the consumption of complement C3, resulting in decreased serum C3 levels. *Tsai, Wu & Chen (2019)* found that almost all patients with aMN had detectable deposits of C3 in renal biopsy tissues, and a higher deposition intensity was associated with lower serum C3 levels, predicting poorer long-term renal outcomes. This study also found that lower serum C3 levels were associated with worse long-term renal prognosis in patients with aMN, consistent with the findings of *Tsai, Wu & Chen (2019)*. Therefore, serum C3 might represent an independent predictive factor or protective factor for long-term renal survival in aMN. Notably, there was no statistically significant correlation between renal tissue complement C3 deposition and poor renal prognosis in aMN in this study, likely due to the limited information provided by the binary classification of tissue complement C3 in the Cox analysis. Serum complement C3, as a quantitative indicator, may indirectly reflect the activation of renal tissue C3 through the decrease in serum C3 levels, which holds greater clinical value in predicting renal prognosis in aMN.

Furthermore, this study found that patients with aMN who had hematuria experienced worse renal outcomes. This was consistent with previously reported clinical findings. *He et al. (2021)* indicated that more severe initial hematuria was predictive of relapse in patients with aMN, with the severity and duration of hematuria being independently related to the progression of renal dysfunction. Infiltrating leukocytes can release metalloproteinases and reactive oxygen species in MN, making the glomerular basement membrane more susceptible to rupture, which cause glomerular hematuria (*He et al., 2021*). Fragmented red blood cells release hemoglobin and iron into renal tubules, causing renal damage through oxidative stress. Hematuria therefore leads to glomerular and tubular damage, contributes to the worse renal outcomes.

Moreover, elevated SCr levels were associated with worse prognoses in patients with aMN. Similarly, *Tsai, Wu & Chen (2019)* indicated that baseline renal dysfunction (high SCr, low eGFR) was a risk factor for progression to ESRD in patients with aMN in univariate analysis. Another retrospective study (*Liu et al., 2022*) involving 1,313 patients with aMN identified baseline SCr as a risk factor for a 50% decline in eGFR through multivariate Cox analysis. In this study, compared with other independent risk factors, SCr had a relatively smaller impact on renal prognosis, suggesting that the renal prognosis of aMN was influenced by multiple factors, and elevated baseline SCr alone could not fully reflect the extent of renal prognosis risk. Therefore, to accurately assess the renal prognosis of patients with aMN, it was necessary to consider multiple indicators comprehensively.

Based on real clinical data, this study performed multivariate Cox regression analysis to identify factors affecting the renal prognosis of patients with aMN and constructed a predictive nomogram. This study had several strengths. First, it examined the differences in renal outcomes among patients with aMN according to the presence of AKI, confirming the role of AKI in poor renal outcomes. Second, it selected common and easily obtainable clinical factors as study variables, making the developed predictive model more practically applicable in actual clinical settings. Lastly, the model exhibited excellent predictive performance for renal prognosis over 3–6 years, helping clinicians to accurately assess the

mid- to long-term renal prognosis of patients with aMN and optimize treatment strategies to improve long-term survival quality.

Despite its meaningful results, this study had certain limitations. First, as a single-center, retrospective analysis with a small sample size, a validation cohort could not be established for external verification, and the use of a consecutive selection method might limit the representativeness of the sample. Therefore, these factors might affect the generalizability of the results. Second, most positive follow-up events occurred after 3 years, which led to insufficient support for short-term (1–2 years) events and restricted the ability to accurately predict short-term renal prognosis for patients with aMN. Additionally, the model exhibited slight overprediction (optimism) in estimating survival probabilities for patients with extremely high predicted survival rates accompanied by increased uncertainty. This phenomenon reflected the potential heterogeneity within this patient group, which might limit the model's predictive ability in extreme cases. Lastly, some variables identified in the final predictive model, including those closely related to aMN remission (*e.g.*, 24-h urine protein), required subgroup analysis and further long-term follow-up to confirm their impact on renal prognosis throughout the course of aMN. Considering the limitations, future research should be conducted with larger sample sizes, a broader patient population, multicenter data, and randomized prospective protocols to further enhance the accuracy, stability, and generalizability of the predictive model.

## CONCLUSION

This study developed a clinical predictive model for renal outcomes in patients with aMN regardless of the presence of AKI. We recommend early assessment and management for related risk factors, especially avoiding risk factors for AKI (*e.g.*, infections, dehydration, and use of nonsteroidal anti-inflammatory drugs) to delay the occurrence of adverse renal outcomes.

## ACKNOWLEDGEMENTS

We acknowledge all patients, investigators, and support staff for their contributions to this research.

### Funding

This study was supported by funding from the Youth Scientific Research Project of Fujian Provincial Health Commission (No. 2021QNA023), the Medical Innovation Project of Fujian Provincial Health Commission (No. 2022CXA025), and the National Natural Science Foundation of China Youth Project (No. 82100740). The funders did not participate in the design of the study, collection or analysis of data, decision to publish, or drafting of the manuscript. The funders had no role in study design, data collection and analysis, decision to publish, or preparation of the manuscript.

## Grant Disclosures

The following grant information was disclosed by the authors:
Youth Scientific Research Project of Fujian Provincial Health Commission: 2021QNA023.
Medical Innovation Project of Fujian Provincial Health Commission: 2022CXA025.
National Natural Science Foundation of China Youth Project: 82100740.

## Competing Interests

The authors declare that they have no competing interests.

## Author Contributions

- Zhenzhou Li conceived and designed the experiments, performed the experiments, analyzed the data, prepared figures and/or tables, authored or reviewed drafts of the article, and approved the final draft.
- Liyan Yang conceived and designed the experiments, performed the experiments, analyzed the data, prepared figures and/or tables, and approved the final draft.
- Linxia Wei conceived and designed the experiments, performed the experiments, analyzed the data, prepared figures and/or tables, and approved the final draft.
- Mengjie Weng conceived and designed the experiments, performed the experiments, prepared figures and/or tables, and approved the final draft.
- Jiaqun Lin performed the experiments, prepared figures and/or tables, and approved the final draft.
- Yi Chen performed the experiments, prepared figures and/or tables, and approved the final draft.
- Binbin Fu performed the experiments, prepared figures and/or tables, and approved the final draft.
- Guifen Li performed the experiments, prepared figures and/or tables, and approved the final draft.
- Caiming Chen performed the experiments, prepared figures and/or tables, and approved the final draft.
- Yanfang Xu performed the experiments, prepared figures and/or tables, and approved the final draft.
- Jianxin Wan conceived and designed the experiments, performed the experiments, authored or reviewed drafts of the article, and approved the final draft.
- Jiong Cui conceived and designed the experiments, performed the experiments, analyzed the data, authored or reviewed drafts of the article, and approved the final draft.

## Human Ethics

The following information was supplied relating to ethical approvals (*i.e.*, approving body and any reference numbers):

This study adhered to all relevant principles of the Declaration of Helsinki and received approval from the Medical Research and Clinical Technology Application Ethics Committee of the First Affiliated Hospital of Fujian Medical University (MTCA, ECFAH of FMU [2015] 084-2).

## Data Availability

The raw data are available in the Supplemental File.

## Supplemental Information

Supplemental information for this article can be found online at http://dx.doi.org/10.7717/peerj.19331#supplemental-information.

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
