# Peer review of "Risk factors and predictive model for renal outcomes in autoimmune membranous nephropathy with and without acute kidney injury: a retrospective cohort study"

_PeerJ, doi:10.7717/peerj.19331_

## Round 0.1 · original submission · Major Revisions

The study methods should be more clearly explained and the overall writing style should be improved.

·

Basic reporting

The authors present a study on idiopathic membranous nephropathy (IMN), particularly examining the impact of acute kidney injury (AKI) on patient prognosis. The research employs extensive data collection spanning 13 years and involving 441 biopsy-proven IMN cases enhances the reliability and representativeness of the findings. Identifying early intervention targets for risk factors like AKI, triglycerides, serum creatinine, hematuria, and PLA2R staining can aid in slowing renal function decline in IMN patients, providing valuable clinical guidance.

Experimental design

The study’s strengths lie in its rigorous design, large sample size, and the development of a predictive model. However, limitations such as single-center bias and variable selection must be addressed to enhance the study’s applicability and robustness. By expanding the scope of future research, the potential impact of this work on clinical practice and patient outcomes can be significantly increased.

Validity of the findings

Major concerns:

1. The research data were derived exclusively from the First Affiliated Hospital of Fujian Medical University, and the regional specificity of the sample may limit the generalizability of the findings. It is recommended to incorporate multicenter data to enhance the generalizability of the results and the reliability of external validation.

2. Despite the large sample size, some subgroups (such as patients with AKI stage 2 and stage 3) had a relatively small number of participants, which may affect the statistical significance of the results. Furthermore, although the follow-up duration was relatively lengthy, there is insufficient support for predicting short-term (1-2 years) outcomes.

3. It is advisable to incorporate additional potential prognostic factors, such as patients' lifestyle, medication adherence, and genetic polymorphisms, in order to comprehensively assess the factors influencing the prognosis of idiopathic membranous nephropathy (IMN) and to enhance the accuracy of the predictive model.

Additional comments

Minor comments:

1.Page 17, line 221-223,The paragraph states that the incidence of IMN in China has gradually increased, while in some western countries it has decreased. However, the explanation for this disparity (i.e., related to lifestyle changes and dietary factors) may lack sufficient evidence or depth. It would benefit from a clearer connection between the identified factors and the observed trends.

2.Page 17, line 233,While the paragraph mentions that few studies have focused on the impact of AKI on renal outcomes in IMN, it does not provide context or rationale for why this is significant. The implications of the lack of research on AKI in this population should be elaborated upon to strengthen the argument for the study's importance.

3. In line 236, "inpatients" should be corrected to "in patients" as it is a spacing error that alters the meaning.

4. Some sentences are quite lengthy and could benefit from being broken down for clarity. For example, the sentence starting with "After treatment for IMN" (lines 223-225) could be divided to enhance readability and comprehension.

5. Page19, The paragraph lacks clear transitions, especially when discussing different risk factors (such as PLA2R and serum triglycerides). It is recommended to add transitional sentences to enhance the logical flow.

6. Page19, While the paragraph mentions that serum triglycerides may lead to renal dysfunction through various mechanisms, the specific causal relationships are not clearly articulated. Readers may find the connection between "lipotoxicity" and "tubular atrophy" confusing.

7. When describing research findings, it is important to maintain consistent tense. Some sentences use the present tense while others use the past tense; it is recommended to use the past tense consistently to describe completed studies.

·

Basic reporting

1. Renal Endpoint: The renal endpoint needs to be explicitly and clearly stated in the abstract for better clarity.
2. Keywords: Key words are missing from the manuscript and should be included.
3. Introduction: In the introduction section, line 55, I suggest removing "hypertension and diabetes" as examples of comorbidities, as these conditions can still coexist with primary membranous nephropathy (PMN).

Experimental design

1. Patient Selection:
o Please clarify whether patients were selected randomly or consecutively. As this is a retrospective study, potential selection bias is important to note.
o I recommend adhering to the STROBE statement, and explicitly addressing its checklist in the methodology section to ensure transparency and robustness.
2. Event Rate and Sample Size:
o With 59 events and 6 predictors in the nomogram, this appears acceptable. However, please clarify how the study's power and required sample size were calculated to ensure robustness.
3. Serum PLA2R Testing:
o While you used indirect IF for serum PLA2R testing, this is a qualitative method. Incorporating EUROIMMUN ELISA, if available, would have provided quantitative data, potentially improving the risk prediction model.
4. Immunosuppression Variable:
o You mention immunosuppression as a categorical variable being statistically significant. Please rerun the model using immunosuppression use (Y/N) as an independent variable and evaluate if this improves the model metrics.
o Note: Adding this as a 7th predictor would require at least 10 more events to ensure statistical reliability as per the "10 events per predictor" rule.
5. Internal Validation:
o Please include in the methodology section that bootstrapping internal validation (e.g., 1000 iterations) was performed, as this is critical for the robustness of the model.
6. Calibration:
o In the discussion, you should comment on the calibration curve, noting that the model shows slight overprediction and wider uncertainty for patients with very high predicted survival probabilities.
7. AKI Definition and Inclusion:
o How was AKI defined for patients presenting with low eGFR and no baseline values or with baseline eGFR measured more than 3 months earlier? These cases might reflect CKD progression rather than AKI.
o If you excluded such patients, make it clear that you only included those with a documented baseline eGFR available on clinical record at presentation.

Validity of the findings

1. Timing of AKI:
o Clarify whether AKI occurred at presentation or later. This is important for understanding the clinical relevance of AKI as a potential predictor.
o For context, our center is working on a risk stratification calculator at presentation. If AKI at presentation is a strong predictor, it could be highly relevant to such tools.
2. Clinical Relevance of Predictors:
o In line 175, you listed predictors of the renal endpoint: AKI, baseline SCr, baseline eGFR, triglycerides, serum C3, hematuria, DBP, high PLA2R expression in the kidneys (4+), chronicity index, and glomerulosclerosis.
 I suggest rephrasing to clearly highlight that AKI, hematuria, and PLA2R (4+) on biopsy are negative predictors and C3 is a positive predictor, as these were the only clinically relevant variables (others had confidence intervals crossing the null hypothesis in the univariable Cox model, although being statisctically significant).
3. Typographical Issue:
o In line 193, you refer to "interval validation." This should likely be "internal validation."
4. Discussion Length:
o The discussion is overly long and should be condensed to improve focus.
5. Serum PLA2R Levels:
o Lines 253–266 discuss serum PLA2R levels, but this section seems irrelevant unless the serum test showed significant results. Since this was a qualitative method, consider removing this discussion.
6. Triglycerides as a Predictor:
o In lines 267–278, you discuss triglycerides as a cause of renal damage. While statistically significant, this variable's confidence interval is close to the null hypothesis in your study, making its clinical relevance questionable. Please highlight this limitation.
7. C3 on Biopsy:
o In lines 279–300, you discuss C3 (serum and biopsy) as predictors. Do you have univariate data for C3 biopsy staining to compare with other studies?
o If not, I suggest removing this section.
o In line 295, you mention higher C3 deposits in IMN and AKI without significance. Please include an interaction model in supplementary material to show which variable is a stronger predictor of the endpoint.

Reviewer 3 ·

Basic reporting

This study explored renal outcomes in patients with IMN with or without AKI and constructed a prognostic predictive model to predict ESRD. It is an interesting study. However, there are some major problems in this study and the writing of this manuscript need improvement. I hope these comments will be helpful for revising your article.

Abstract
What are the main endpoints of the study? The methods section of the abstract needs to be clarified, and the rest of the writing should be simplified.

Introduction
1. The percentage of MN entering ESRD is associated with the follows-up duration. How long of follow-up the percentage 20% represented?
2. Although the studied risk factors can predict prognosis, it cannot be determined whether early intervention can improve prognosis, and there are no studies suggesting that early intervention can improve prognosis.
3. Hypertension and diabetes are not secondary causes of MN.

Experimental design

Methods
1. The selected patients are solely those who underwent kidney biopsy. For membranous nephropathy, unless there are exceptional circumstances, the future trend is that renal biopsies will become less common, and including only patients who have undergone kidney biopsies will narrow the applicability of the study results.
2. FSGS is excluded in the exclusion criteria; generally, membranous nephropathy is not considered to be combined with primary FSGS, although segmental sclerosis may present pathologically in cases of MN. If the latter, I don’t think these FSGS cases need to be excluded.
3. Why was only C3 selected for testing, and not C4?
4. Subjects who have elevated creatinine upon admission and then show a decline in creatinine after admission are also defined as having experienced AKI. By how much must creatinine decrease after admission relative to the initial admission value to be defined as AKI? How will AKI grading be conducted for this group of subjects? What proportion do these subjects represent among the 109 cases with AKI? For patients with urine output less than 0.5 ml/kg/hour, since patients with membranous nephropathy are typically not critically ill, it is unlikely that hourly urine output will be accurately recorded; how can a precise judgment be made?
5. Where does the source for the composite endpoint of a decrease in eGFR of >40% from baseline come from? Why is this definition used? Death is the hardest endpoint; why is it classified as censored? The composite endpoint mentions the initiation of dialysis; if temporary replacement therapy is used due to AKI, does this count as a primary endpoint event?

Validity of the findings

Result
1. Does AKI refer to the occurrence of AKI at the time of consultation, or does it include any AKI during the follow-up process? AKI patients often have higher creatinine levels, and if the timing of AKI coincides with the baseline, the two are not independent and may not be appropriate to include in the same model.
2. PLA2R + up to 4+ is an ordinal variable representing different degrees of the same indicator and should be treated as a single variable for statistical analysis.
3. The immunosuppressive regimen is a key factor affecting prognosis. It is recommended to provide specific details about the immunosuppressive regimen, as well as whether immunosuppressive treatment and its protocols impact prognosis.
4. There are significant differences in baseline SCr, UP, and Alb, and these factors are related to the occurrence of AKI. Whether patients reach the endpoint may be related to these indicators, so it is suggested to conduct a multivariate analysis to include these factors, as they seem to be more important in influencing renal prognosis.
5. In the univariate Cox regression analysis, there are 10 variables with p < 0.05; why were DBP, eGFR, Chronicity index, and Glomerular sclerosis not included in the multivariate Cox regression analysis?
6. “...some patients had elevated SCr levels on the first day of admission, followed by a decline. The baseline SCr levels of these patients were assumed to be lower than those on the first day based on the subsequent clinical course. Thus, based on the 2012 KDIGO Clinical Practice Guidelines for AKI, these patients were also diagnosed with AKI.”
(1) How many patients fall into this category?
(2) By how much did the creatinine levels of these patients decrease, and was there a specific range limit for the definition?
It is recommended to supplement detailed data for this group of patients.
7. Table 2 shows that 93% of subjects have IgG deposition; how is MN diagnosed in the remaining 7% if no IgG deposition is observed?
8. Why was Kidney PLA2R negative not included in the univariate Cox analysis?
9. There is a lack of external cohort validation after the predictive model was established.
10. Line 160: Is the writing of quartiles not standardized? “(IQR = 43, 63)” should be written as “(IQR = 43–63)” or “(IQR: 43–63).” Lines 158–165: The decimal places for percentages within parentheses should be uniform.

Additional comments

No

---

## Round 0.2 · Major Revisions

Please review the final comment of reviewer 2 and send your response.

·

Basic reporting

I would suggest changing IMN (idiopathic membranous nephropathy) throughout the manuscript to aMN (autoimmune Membranous nephropathy) as we now know that it is not idiopathic anymore and it is driven by an autoimmune phenomenon which could be PLA2R,NELL1,EXO1,THSD7A ETC..

Experimental design

.

Validity of the findings

.

Additional comments

.

Reviewer 3 ·

Basic reporting

No comment

Experimental design

No comment

Validity of the findings

No comment

Additional comments

No additional comments

---

## Round 0.3 · accepted · Accept

All the previous suggestions were resolved by the authors. No further comments.